# LEARNING DISEASE PROGRESSION MODELS THAT CAPTURE HEALTH DISPARITIES

## ABSTRACT

Disease progression models are widely used to inform the diagnosis and treatment of many progressive diseases. However, a significant limitation of existing models is that they do not account for health disparities that can bias the observed data. To address this, we develop an interpretable Bayesian disease progression model that captures three key health disparities: certain patient populations may (1) start receiving care only when their disease is more severe, (2) experience faster disease progression even while receiving care, or (3) receive follow-up care less frequently conditional on disease severity. We show theoretically and empirically that failing to account for disparities produces biased estimates of severity (underestimating severity for disadvantaged groups, for example). On a dataset of heart failure patients, we show that our model can identify groups that face each type of health disparity, and that accounting for these disparities meaningfully shifts which patients are considered high-risk.

## 1 INTRODUCTION

In many settings, observed data is used to model the progression of a latent variable over time. Models of human aging use a person's physical and biological characteristics to model progression of their latent "biological age" (Pierson et al., 2019); models of infrastructure deterioration use inspection results to model progression of a system's latent overall health (Madanat et al., 1995); and disease progression models, which we focus on in this paper, use observed symptoms to model progression of a patient's latent severity of a chronic disease (Wang et al., 2014). Disease progression models can help predict a patient's disease trajectory and thus personalize care, detect diseases at earlier stages, and guide drug development and clinical trial design (Mould et al., 2007; Romero et al., 2015). They have been applied to a wide variety of progressive diseases such as Alzheimer's disease (Holford & Peace, 1992) and cancer (Gupta & Bar-Joseph, 2008).

For the benefits of these models to apply to all patients equitably, it is crucial that they accurately describe progression for all populations of patients. However, disease progression models have typically failed to account for the fact that systemic disparities in the healthcare process can bias the observed data that they are trained on. For example, disparities have been shown to arise along axes such as socioeconomic status (Weaver et al., 2010; Miller & Wherry, 2017), race (Yearby, 2018), and proximity to care (Chan et al., 2006; Reilly, 2021). Accounting for such disparities is important because it can meaningfully shift estimates of disease progression. For intuition, imagine learning that a patient in the emergency room traveled three hours to get to the hospital; if their symptoms are ambiguous, this contextual information may increase our estimate of how severe their underlying condition is. Disease progression models have historically been unable to capture this type of social context—as we show later, this can lead to biased estimates of severity. To address this, we propose a method for learning disease progression models that interpretably capture three well-documented health disparities:

1. **Disparities in initial severity.** Certain patient groups may start receiving care only when their disease is more severe (Hu et al., 2024).

2. **Disparities in disease progression rate.** Certain patient groups may experience faster disease progression, even while receiving care (Diamantidis et al., 2021).

3. **Disparities in visit frequency.** Certain patient groups may visit healthcare providers for follow-up care less frequently, even at the same disease severity (Nouri et al., 2023).

It is a core technical challenge to design a model that is flexible enough to capture all three disparities but still identifiable. Identifiability is necessary for accurate estimates of disparities and disease progression. As such, our key contributions are: (1) we develop an interpretable Bayesian model of disease progression that accounts for multiple types of disparities but remains provably identifiable from the observed data; (2) we prove and show empirically that failing to account for any of these three disparities leads to biased estimates of severity; and (3) we characterize fine-grained disparities in a heart failure dataset. Our model reveals that non-white patients have more severe heart failure and face multiple types of health disparities: Black and Asian patients tend to start receiving care at more severe stages of heart failure than do White patients, and Black patients see healthcare providers for heart failure 10% less frequently than do White patients at the same disease severity level. Accounting for these disparities meaningfully shifts our estimates of disease severity, increasing the fraction of non-white patients identified as high-risk. While we ground our work in healthcare, our method for learning progression models that account for disparities applies naturally to many other progression model settings where disparities are of interest, including infrastructure deterioration (Madanat et al., 1995) and human aging (Pierson et al., 2019).

## 2 RELATED WORK

**Disease progression modeling.** Disease progression models have been developed for many chronic diseases, including Parkinson's disease (Post et al., 2005), Alzheimer's disease (Holford & Peace, 1992), diabetes (Perveen et al., 2020), and cancer (Gupta & Bar-Joseph, 2008). A key feature of the progression models we consider, common in the machine learning literature, is that a latent severity $Z_t$ progresses over time and gives rise to the observed symptoms $X_t$. Models in this family include variants of hidden Markov models (HMMs) (Wang et al., 2014; Liu et al., 2015; Alaa & Hu, 2017; Sukkar et al., 2012; Jackson et al., 2003) and recurrent neural networks (RNNs) (Choi et al., 2016b; Lipton et al., 2017; Lim & van der Schaar, 2018; Choi et al., 2016a; Ma et al., 2017; Kwon et al., 2019; Alaa & van der Schaar, 2019). The existing literature has not focused on modeling disparities; we extend it by proposing a new approach to disease progression modeling that can interpretably characterize and account for multiple types of health disparities.

**Health disparities.** Disparities have been documented in many parts of the healthcare process. Factors such as distance from hospitals (Reilly, 2021), distrust of the healthcare system (LaVeist et al., 2009), or lack of insurance (Venkatesh et al., 2019) can result in underutilization of health services; biases in the judgements of healthcare providers can lead minority groups to receive later screening (Lee et al., 2021), fewer referrals (Landon et al., 2021), or generally worse care (Schäfer et al., 2016); and issues such as limited health literacy or trust can create disparities in follow-through for appointments or the effectiveness of at-home care (Davis, 1968; Brandon et al., 2005).

The existing literature has shown that disparities emerge along the three axes that we capture in this paper: (1) how severe a patient's disease becomes before they start to receive care (Chen et al., 2021; Iqbal et al., 2015; Hu et al., 2024); (2) how quickly their latent severity progresses even while receiving care (Diamantidis et al., 2021; Suarez et al., 2018); and (3) how likely they are to visit a healthcare provider at a given severity level (Nouri et al., 2023). Our goal is to show how accounting for disparities along all three of these axes improves the severity estimates of disease progression models, while also learning more fine-grained descriptions of existing disparities.

**Capturing disparities with machine learning.** We build upon a large body of past work that uses machine learning as a tool to capture and address health disparities, including models that estimate the relative prevalence of underreported medical conditions (Shanmugam et al., 2021), improve risk prediction for patients with missing outcome data (Balachandar et al., 2023), evaluate the impact of race corrections in risk prediction (Zink et al., 2023), assess disparate impacts of AI in healthcare (Chen et al., 2019), and quantify disparities in the performance of clinical prediction tasks (Zhang et al., 2020). The closest work to our own is Chen et al. (2021), which develops a clustering algorithm that accounts for the fact that some patients do not come in (and are therefore not observed) until later in their disease progression. While their work addresses one form of data bias that can arise due to health disparities, it differs from our own in two ways: it does not specifically document or study health disparities, and it focuses on clustering patients as opposed to modeling disease severity or progression. Our work proposes a model for capturing three types of health

disparities in the disease progression setting in order to learn precise descriptions of multiple disparities and make severity estimates that exhibit less bias than existing disease progression models.

## 3 MODEL

We build on a standard setup for disease progression modeling, in which each patient has an underlying latent disease severity $Z_t$ that progresses over time and gives rise to a set of observed features $X_t$ (Klemera & Doubal, 2006; Levine, 2013).

We characterize each patient's severity $Z_t \in \mathbb{R}$ at time $t$ by their *initial severity* $Z_0$ at their first observation (which we denote as $t = 0$) and their *rate of progression* $R$ after that point:

$$Z_t = Z_0 + R \cdot t$$

If a patient visits a healthcare provider at time $t$, we observe some recorded set of *features* $X_t \in \mathbb{R}^d$ (e.g., lab results, imaging, symptoms). At any given visit, a clinician does not necessarily observe or record all features—we model the features that *are* observed as a noisy function of their latent severity $Z_t$:

$$X_t = f(Z_t) + \epsilon_t$$
$$\epsilon_t \sim \mathcal{N}(0, \Psi)$$

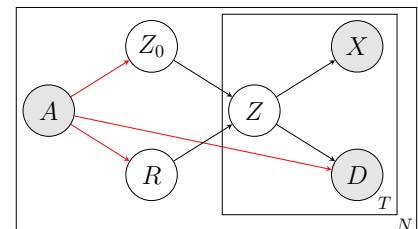

Figure 1: **Disease progression generative model.** Plate diagram captures $N$ patients over $T$ timesteps. Shaded nodes indicate observed features: demographics $A^{(i)}$, visit indicator $D_t^{(i)}$, and symptoms $X_t^{(i)}$ (only observed when $D_t^{(i)} = 1$). Unshaded nodes indicate latent variables: a patient's initial severity $Z_0^{(i)}$, rate of progression $R^{(i)}$, and severity $Z_t^{(i)}$. Red arrows indicate dependencies capturing health disparities.

where the diagonal covariance matrix $\Psi \in \mathbb{R}^{d \times d}$ parameterizes feature-specific noise (accounting for both measurement error and variation in how the patient's physical state can fluctuate day-to-day). In our experiments, we specifically instantiate $f$ as a linear function $f(Z_t) = F \cdot Z_t + b$, where $F \in \mathbb{R}^d$ is a feature-specific scaling factor and $b \in \mathbb{R}^d$ is a feature-specific intercept, but our approach extends to more general parametric forms for $f$. We constrain the first feature $F_0 > 0$ using domain knowledge; this restriction is necessary for identifiability because it restricts the mapping between features and severity (Shapiro, 1985). We also observe a set of timesteps when a patient visits a healthcare provider; we discretize time and indicate whether a patient visits a healthcare provider at time $t$ with a binary indicator $D_t \in \{0, 1\}$.

**Capturing disparities.** Our model captures the three types of health disparities discussed in §2 by allowing model parameters to vary as a function of a patient's demographic feature vector $A$ (Figure 1). For expositional clarity, we describe a setup where $A$ encodes a single categorical label (e.g., a patient's race group), but our approach naturally extends to multiple categorical groupings or to continuous features.

1. **Disparities in initial severity.** Underserved patients may start receiving care only when their disease is more severe. We capture this by learning group-specific distributions of $Z_0$, a patient's disease severity at their first visit. For one group $A = a_0$, we pin $Z_0$ to be drawn from a unit normal distribution; this is a standard and necessary identifiability condition since it fixes the scale of $Z_t$ (Shapiro, 1985). For other groups $a$,

$$Z_0 \sim \mathcal{N}\left(\mu_{Z_0}^{(a)}, \sigma_{Z_0}^{(a)^2}\right)$$

where $\mu_{Z_0}^{(a)}$ and $\sigma_{Z_0}^{(a)}$ are learned group-specific parameters.

2. **Disparities in disease progression rate.** Underserved patients may experience faster disease progression even while receiving care. We capture this by learning group-specific distributions of disease progression rate $R$:

$$R \sim \mathcal{N}\left(\mu_R^{(a)}, \sigma_R^{(a)^2}\right)$$

where $\mu_R^{(a)}$ and $\sigma_R^{(a)}$ are learned group-specific parameters for each group $a$.

3. **Disparities in visit frequency.** Underserved patients may visit healthcare providers for follow-up care less frequently at the same disease severity. We capture this by modeling patient visits as generated by an inhomogeneous Poisson process, parameterized by a time-varying rate parameter $\lambda_t$ that depends on both $Z_t$ and $A$:

$$\log(\lambda_t) = \beta_0 + \beta_Z \cdot Z_t + \beta_A^{(a)}$$

where $\beta_Z$ and $\beta_0$ are learned parameters for the entire population and $\beta_A^{(a)}$ is a learned group-specific parameter for each group $a$ (we pin $\beta_A^{(a_0)} = 0$ for reference).

Overall, our model parameters (on which we place weakly informative priors) are the parameters shared across groups $\{F, b, \Psi\}$, and the group-specific parameters $\{\mu_{Z_0}^{(a)}, \sigma_{Z_0}^{(a)}, \mu_R^{(a)}, \sigma_R^{(a)}, \beta_0, \beta_Z, \beta_A^{(a)}\}$. We learn posterior distributions over these parameters from our observed data $X_t, D_t, A$ using Hamiltonian Monte Carlo, a standard algorithm for Bayesian inference (Betancourt, 2018), as implemented in Stan (Carpenter et al., 2017). Figure 1 summarizes the data generating process and Table 1 summarizes the notation for our model.

**Model discussion.** Modeling progression as linear over time is a common approach (Holford & Peace, 1992; Pierson et al., 2019), because it provides an interpretable characterization of the trajectory. The interpretability of using a single intercept and progression rate parameter to characterize a patient's disease trajectory is especially valuable in our setting, allowing us to compare how severe groups are at initial presentation and how quickly they progress. Similarly, using a Poisson process to model event frequency is a common approach, including in work that seeks to capture disparities in event frequency (Liu et al., 2024; Kurashima et al., 2018).

| Notation | Meaning |
|----------|---------|
| $X_t$ | Observed features at time $t$ |
| $D_t$ | Binary visit indicator for time $t$ |
| $A$ | Demographic features |
| $Z_t$ | Disease severity at time $t$ |
| $Z_0$ | Initial severity |
| $R$ | Disease progression rate |
| $F$ | Severity-feature matrix |
| $b$ | Feature intercepts |
| $\Psi$ | Feature covariance matrix |
| $\mu_{Z_0}, \sigma_{Z_0}$ | Group-specific mean and sd of $Z_0$ |
| $\mu_R, \sigma_R$ | Group-specific mean and sd of $R$ |
| $\lambda_t$ | Visit rate at time $t$ |
| $\beta_0$ | Visit rate intercept |
| $\beta_Z$ | Visit rate $Z_t$ coefficient |
| $\beta_A$ | Visit rate $A$ coefficient |

Table 1: **Summary of notation.** Observed data are listed above the double horizontal line.

## 4 THEORETICAL ANALYSIS

In this section, we prove two main theoretical results. First, we show that our model is *identifiable*, a necessary condition for its parameters to be estimated from the observed data and interpreted. Learning these parameter estimates is what allows us to characterize disparities. Second, we prove that failing to account for disparities produces *biased estimates of severity*. We summarize proof strategies in the main text and provide formal proofs in Appendices §A and §B.

### 4.1 IDENTIFIABILITY

We show that our model is identifiable, meaning different sets of parameters yield different observed data distributions (Bellman & Åström, 1970) and thus that we can recover correct estimates of all model parameters from the observed data. Learning a model of progression that is *flexible* enough to characterize multiple disparities but *still identifiable* is a fundamental challenge. In fact, if we added one more dependence on $A$ — in particular, adding an arrow from $A$ to $X$ in Figure 1 — the model would no longer be identifiable; without a shared interpretation across groups of how features map to severity, it would be impossible to identify disparities in disease progression. Put another way, our model encodes the richest set of disparities on the observed data while retaining identifiability.

**Theorem 4.1.** *All model parameters are identified by the observed data distribution $P(X_t, D_t \mid A)$.*

As mentioned in §3, the distribution of initial severity $Z_0$ is pinned to a unit normal for one demographic group $a_0$. This pinned distribution reduces the number of unknown latent parameters for group $a_0$, allowing us to show that $\{F, b, \Psi\}$ are identified by $P(X_t \mid A = a_0)$. Having identified

these, we show that the parameters $\{\mu_{Z_0}^{(a)}, \sigma_{Z_0}^{(a)}, \mu_R^{(a)}, \sigma_R^{(a)}\}$ are identified by $P(X_t \mid A = a)$ for all groups $a$. Finally, we show that given the previously identified parameters, $\{\beta_0, \beta_Z\}$ are identified by $P(D_t \mid A = a_0)$ and $\{\beta_A^{(a)}\}$ is identified by $P(D_t \mid A = a)$ for all other groups $a$.

## 4.2 BIAS IN MODELS THAT DO NOT ACCOUNT FOR DISPARITIES

Next we show that, when any of the health disparities we discuss are present, a model that does not account for group-specific disparities will produce *biased estimates* of severity—i.e., $\mathbb{E}[Z_t \mid X_t, D_t] \neq \mathbb{E}[Z_t \mid X_t, D_t, A = a]$. These theoretical results hold under more general assumptions than our full parametric model: our assumptions, which we formally describe in Appendix B, are that the model dependencies are encoded by the DAG in Figure 1; that severity $Z_t$ increases linearly with progression rate $R$; and that visit rate $\lambda_t$ increases with severity $Z_t$. For each proof, we analyze the effect of one disparity — e.g., for disparities in initial severity, we assume that $P(Z_0 \mid A = a)$ differs across groups — while keeping other distributions constant across groups. These results hold in the presence of multiple disparities as long as existing disparities disfavor or favor the same group, so as to not cancel each other out in their effects.

We quantify disparities by using the strict Monotone Likelihood Ratio Property (MLRP) to reason about the probability density functions of initial severity and progression rate for certain groups, relative to the overall population (Karlin & Rubin, 1956):

**Definition 4.2.** Two distributions characterized by probability density functions $f(x)$ and $g(x)$ have the strict monotone likelihood ratio property in $x$ if $\frac{f(x)}{g(x)}$ is a strictly increasing function of $x$.

Intuitively, this means that as some variable $x$ ($Z_0$ or $R$, in our case) gets larger, it is more likely to be drawn from $f$ than $g$. The MLRP is a widely-used assumption across many settings (Gaebler & Goel, 2024; Anwar & Fang, 2006; Chemla & Hennessy, 2019); the normal, exponential, binomial, and Poisson families all have this property. For brevity, we say "$f(x)$ strictly MLRPs $g(x)$" to mean that $f(x)$ and $g(x)$ satisfy the strict MLRP in $x$. We now prove for each disparity that any model that fails to account for the disparity will produce biased estimates of severity.

**Theorem 4.3.** *A model that does not take into account disparities in initial disease severity $Z_0$ will underestimate the disease severity of groups with higher initial severity and overestimate that of groups with lower initial severity. Specifically, if $P(Z_0 \mid A = a)$ strictly MLRPs $P(Z_0)$ for some group $a$, then $\mathbb{E}[Z_t \mid X_t] < \mathbb{E}[Z_t \mid X_t, A = a]$. Similarly, if $P(Z_0)$ strictly MLRPs $P(Z_0 \mid A = a)$ for some group $a$, then $\mathbb{E}[Z_t \mid X_t] > \mathbb{E}[Z_t \mid X_t, A = a]$.*

We prove this by showing that $P(Z_0 \mid X_t, A = a)$ strictly MLRPs $P(Z_0 \mid X_t)$, which implies that $\mathbb{E}[Z_t \mid X_t, A = a] > \mathbb{E}[Z_t \mid X_t]$; §B.1 provides a full proof.

**Theorem 4.4.** *A model that does not take into account disparities in rate of progression $R$ will underestimate the disease severity of groups with higher progression rates and overestimate that of groups with lower progression rates. Specifically, if $P(R \mid A = a)$ strictly MLRPs $P(R)$ for some group $a$, then $\mathbb{E}[Z_t \mid X_t] < \mathbb{E}[Z_t \mid X_t, A = a]$. Similarly, if $P(R)$ strictly MLRPs $P(R \mid A = a)$ for some group $a$, then $\mathbb{E}[Z_t \mid X_t] > \mathbb{E}[Z_t \mid X_t, A = a]$.*

We use a similar proof technique as for Theorem 4.3 and provide a full proof in §B.2.

**Theorem 4.5.** *A model that does not take into account disparities in visit frequency $\lambda_t$ (conditional on disease severity) will underestimate the disease severity of groups with lower visit frequency and overestimate that of groups with higher visit frequency. Specifically, if it holds for some group $a$ that $\beta_A^{(a)} < \beta_A^{(\tilde{a})}$ for all $\tilde{a} \neq a$, then $\mathbb{E}[Z_t \mid D_t] < \mathbb{E}[Z_t \mid D_t, A = a]$. Similarly, if it holds for some group $a$ that $\beta_A^{(a)} > \beta_A^{(\tilde{a})}$ for all $\tilde{a} \neq a$, then $\mathbb{E}[Z_t \mid D_t] > \mathbb{E}[Z_t \mid D_t, A = a]$.*

Since group-specific differences in visit rate at a given severity are captured directly by the $\beta_A$ parameter, we reason about disparities by comparing these parameters by group. We prove the theorem by directly reasoning about the estimates of $Z_t$ when considering the additional term $\beta_A$ versus not, reasoning in the large-sample limit in which $\lambda_t$ can be perfectly estimated from the observed data $D_t$. In §5 we show empirically that our results hold in finite samples as well. Overall, these results convey the importance of accounting for disparities in disease progression models: it is fundamentally not possible to make well-calibrated estimates of severity without accounting for group differences in initial severity, progression rate, and visit frequency.

## 5 SYNTHETIC EXPERIMENTS

In this section, we validate our model and theoretical results in synthetic data simulations. We generate synthetic datasets according to the modeling assumptions in §3 (with parameter values for each dataset drawn randomly from each parameter's prior distribution). For each dataset, we generate simulated data for two separate groups, differing in initial severity, progression rate, and visit frequency (characterized by different $\mu_{Z_0}$, $\mu_R$, and $\beta_A$, respectively).

### 5.1 IDENTIFIABILITY AND SEVERITY ESTIMATION

We first verify Theorem 4.1 in simulations, showing that when we fit our model on synthetic data, it can accurately recover the true data-generating parameters. We do this by examining the concordance between the model's estimated parameters and the true, latent parameter values, a common approach in past work (Chang et al., 2021; Pierson et al., 2019). We find high correlation between the true parameters and our model's posterior mean estimates (mean Pearson's $r$ 0.996 across all parameters; median 0.998), and good calibration (mean linear regression slope 1.00; median 1.00 when fit without an intercept term). We provide scatterplots of the true and estimated parameters in Appendix C. We also see that our model's mean *severity estimates* for each group are highly correlated and well-calibrated with ground truth, despite underlying differences in group severity distributions and visit rates (Figure 2).

### 5.2 BIAS IN MODELS THAT DO NOT ACCOUNT FOR DISPARITIES

We now demonstrate in simulation that failing to account for disparities can lead to biased severity estimates, consistent with Theorems 4.3, 4.4, and 4.5. In each trial, we use the same data to fit four models: our full model, which accounts for all disparities, plus three ablated models that each fail to account for one of the disparities (initial severity, progression rate, visit frequency). To characterize the resulting bias of failing to account for each type of disparity, we compute the average error in severity estimates (mean inferred estimate minus mean true severity) of each model, broken down by group. For each ablated model and trial, we define the "underserved group" to be the one that is underserved with respect to the specific disparity that the model fails to capture. When evaluating our full model, we define the "underserved group" to be the one with higher initial severity.

As seen in Table 2, the models that do not account for disparities produce biased estimates: while our full model achieves average error across all trials $-0.02$ and 0 for underserved and other patient groups respectively, the ablated models all have negative error for underserved patients (underestimated severity) and positive error for other patients (overestimated severity). The ablated models also produce severity estimates that are less correlated with true severity.

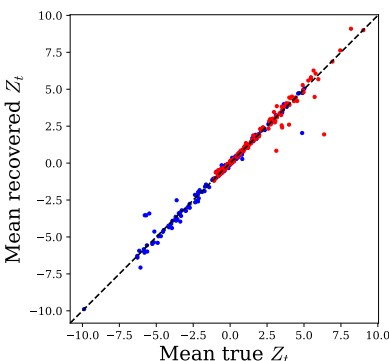

Figure 2: **Well-calibrated severity estimates.** Each dot shows the mean true vs. mean recovered severity values for one group in a given simulation trial. Members of groups depicted in red tend to be underserved compared to groups depicted in blue. Our full model produces accurate and well-calibrated severity estimates (estimates lie near dotted $y = x$ line).

## 6 MODELING HEALTH DISPARITIES IN HEART FAILURE PROGRESSION

We fit our model on a real-world dataset of heart failure patients in the New York-Presbyterian hospital system. Heart failure is a progressive disease that affects many people, requires both specialty and preventive care (Colucci et al., 2020), and has known health disparities (Lewsey & Breathett, 2021), making it a natural application setting for our model. In §6.1 we summarize the dataset, and in §6.2 we confirm that our model can learn meaningful low-dimensional representations of disease severity by evaluating its reconstruction and predictive performance compared to standard baselines. In §6.3 we present our main results: we interpret our model's learned parameters to provide precise

| | Full model | Model that fails to account for disparities in... | | |
|---|---|---|---|---|
| | | Initial severity | Progression rate | Visit frequency |
| Underserved group bias | -0.02 | -0.78 | -0.24 | -0.88 |
| Non-underserved group bias | 0 | +1.03 | +0.01 | +0.42 |
| Underserved group correlation | 0.98 | 0.72 | 0.93 | 0.94 |
| Non-underserved group correlation | 0.99 | 0.69 | 0.94 | 0.93 |

Table 2: **Failing to account for disparities produces biased estimates of severity** $Z_t$**.** We compare severity estimates from our full model to three ablated models that each fail to account for one of the three health disparities. While our full model produces accurate, well-calibrated severity estimates, each ablated model underestimates severity for the underserved group and overestimates it for the other group. The ablated model estimates are also *less correlated* with the true severity values.

descriptions of health disparities in our setting, and we show that (as our theory predicts) failing to account for these disparities meaningfully shifts severity estimates.

## 6.1 DATA

Our data comes from the New York-Presbyterian (NYP)/Weill Cornell Medical Center's electronic health record (EHR) system from 2012 - 2020. We analyze a cohort of $N = 2,942$ patients who (1) have a specific subtype of heart failure (heart failure with reduced ejection fraction), to ensure our cohort can be described by a single progression model, and (2) are likely to receive most of their cardiology care in the NYP system, to ensure we can reasonably estimate when they receive care.

Observed feature data $X_t$ for each patient includes four types of measurements: left ventricle ejection fraction (LVEF), brain natriuretic peptide (BNP), systolic blood pressure (SBP), and heart rate (HR). LVEF and BNP have strong clinical associations with heart failure severity (in terms of both underlying physiological health and observed symptoms) (Murphy et al., 2020). SBP and HR are less informative (more prone to fluctuation and changes not related to heart failure), but they are still expected to show general trends over time as a patient's heart failure progresses. Since we must pin the sign of at least one scaling factor $F$ for identifiability, and decreasing LVEF is strongly associated with increasing severity in the heart failure subtype we study, we pin the sign of the scaling factor between severity and LVEF values ($F_{\text{LVEF}} < 0$).

We discretize time into 1-week bins and observe timesteps when patients receive care. We then analyze disparities across four self-reported race/ethnicity groups: White non-Hispanic patients, Black non-Hispanic patients, Hispanic patients, and Asian non-Hispanic patients (which we will hereby describe as White, Black, Hispanic, and Asian subgroups). A full description of our data processing can be found in Appendix D.

## 6.2 MODEL VALIDATION

We first confirm that our model accurately fits the data: we verify that the model's inferred parameters are consistent with medical knowledge (§6.2.1) and compare the model's reconstruction and predictive performance to standard baselines (§6.2.2). Having confirmed this, we then show in §6.3, as our primary result, that our model provides insight into disparities in disease progression.

### 6.2.1 CONSISTENCY WITH MEDICAL KNOWLEDGE

Figure 3 plots our model's inferred parameters, all of which are consistent with existing medical knowledge.[1] Specifically, (1) the model correctly learns that BNP and HR tend to increase with heart failure severity ($F_{\text{BNP}}, F_{\text{HR}} > 0$), while SBP tends to decrease ($F_{\text{SBP}} < 0$) (Murphy et al., 2020); (2) the model learns larger variance parameters for SBP and HR values ($\psi$), correctly inferring that

---

[1]For succinctness, Figure 3 plots only the model parameters of primary interest for interpreting our model (omitting, for example, estimated intercepts for each feature); a similar coefficient plot with all learned parameters is shown in Figure S2.

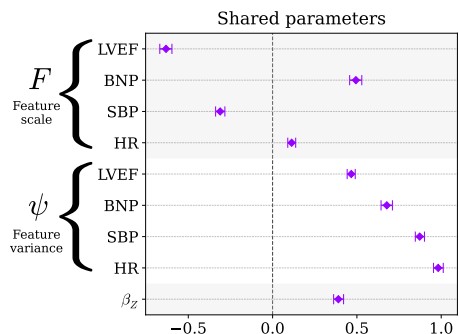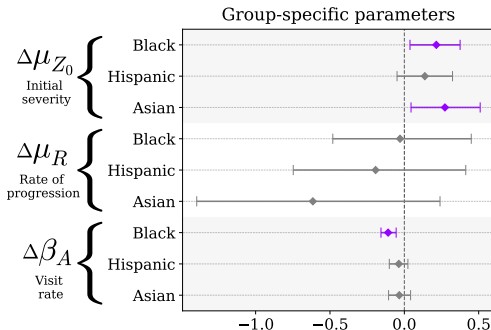

Figure 3: **Inferred model parameters with 95% confidence intervals.** Shared parameters (left) are consistent with medical knowledge of heart failure progression. Group-specific parameters (right) are plotted as differences compared to White patients, so confidence intervals that are non-overlapping with 0 (colored in purple) indicate significant racial/ethnic differences in parameters.

these features are less informative about heart failure progression than are BNP and LVEF (Murphy et al., 2020); and (3) the model estimates that $\beta_Z$ is positive, meaning it learns that patients with higher disease severity tend to see healthcare providers more frequently, as expected.

### 6.2.2 RECONSTRUCTION AND PREDICTIVE PERFORMANCE

We next evaluate the model's ability to reconstruct and predict patient features $X_t$. Because the model represents each patient visit in terms of a scalar severity $Z_t$, we do not expect the model to perfectly reconstruct the multi-dimensional $X_t$; rather, we hope for predictions that correlate significantly with $X_t$. Consistent with this, when fit on 3 years of data per patient, our model's predicted feature values correlate with true values both in- and out-of-sample. As we would hope, the model best represents the features that are most informative for heart failure progression—LVEF ($r = 0.81$ in-sample, $r = 0.51$ out-of-sample) and BNP ($r = 0.62$ in-sample, $r = 0.31$ out-of-sample)—as opposed to the less-informative features SBP ($r = 0.42$ in-sample, $r = 0.24$ out-of-sample) and HR ($r = 0.17$ in-sample, $r = 0.03$ out-of-sample; all p-values besides HR out-of-sample $< 0.001$).

To provide a more detailed assessment of performance, we evaluate our model's ability to *reconstruct* features $X_t$ in-sample and *predict* $X_t$ out-of-sample, in comparison to seven standard baselines. All of the baselines are designed to reconstruct or predict observed feature values ($X_t$), as opposed to additionally predicting whether patient visits will occur ($D_t$). Our model can predict the latter as well, but in order to provide a direct comparison of reconstruction and predictive performance, we compare only the feature prediction aspect of our model (so we do not fit any models using $D_t$ data) in this subsection. In the main text we report mean absolute percentage error (MAPE) of estimated feature values because it allows us to report a normalized measure of error across multiple feature values; in Appendix E we additionally report RMSE.

**Reconstruction performance.** We compare our model's reconstruction performance to that of two standard *dimensionality reduction baselines*: principal component analysis (PCA) and factor analysis (FA). We compare our model to two variants of each. First, we compare our model to PCA and FA fit at the *visit level*: one component per patient visit, analogous to our model's $Z_t$. Second, we compare our model to PCA and FA fit at the *patient level*: two components for each patient, to capture the trajectory of feature values as we do with $Z_0$ and $R$. We describe the implementation of these baselines with more detail in Appendix E.

Because both PCA and FA require input vectors of consistent size, all models are fit on feature values from the first three visits per patient. In Table 3, we report MAPE values averaged across all features as well as across just the more informative features for heart failure severity: LVEF and BNP. We achieve equivalent or better reconstruction performance across all features, and we reconstruct the more informative features more accurately than any of the baselines.

| | Our model | FA$_{visit}$ | PCA$_{visit}$ | FA$_{patient}$ | PCA$_{patient}$ |
|---|---|---|---|---|---|
| MAPE: informative | 20% | 28% | 23% | 25% | 21% |
| MAPE: all | 16% | 19% | 17% | 18% | 16% |

Table 3: **Our model compared to standard baselines for reconstruction performance.** We compare to factor analysis and principal component analysis fit at the patient visit level (FA$_{visit}$, PCA$_{visit}$) and at the trajectory level (FA$_{patient}$, PCA$_{patient}$). Models are fit on the first 3 visits from each patient and evaluated on same data using mean absolute percentage error (MAPE).

| | Our model | Linear regression | Quadratic regression | Latest timestep |
|---|---|---|---|---|
| MAPE: informative | 28% | 39% | 59% | 22% |
| MAPE: all | 21% | 32% | 49% | 18% |

Table 4: **Our model compared to standard baselines for predictive performance.** We compare to linear regression, quadratic regression, and latest timestep prediction, each fit at the patient feature level. Models are fit on data from the first 3 years of each patient's disease trajectory and evaluated on visits after 3 years using mean absolute percentage error (MAPE).

**Predictive Performance.** We also compare our model's predictive performance to that of three standard *timeseries forecasting baselines*: (1) a linear regression for each patient and feature; (2) a quadratic regression for each patient and feature; and (3) predicting values equal to those at the last timestep in training data. For this comparison, all models are fit on feature values from the first three years of data per patient, and we evaluate predictive performance on all remaining visits. As seen in Table 4, our model outperforms both linear regression and quadratic regression on all features. Our model has slightly higher error than latest timestep, which is a widely-used, strong baseline for pure predictive performance (Hyndman, 2018); latest timestep does not, however, provide any insight into disparities or even patterns of progression over time.

Overall, while predicting and reconstructing $X_t$ is not the primary goal of our model, it performs generally well relative to standard baselines, validating its ability to accurately represent the data.

### 6.3 ANALYSIS OF DISPARITIES

We now discuss three main findings from fitting our model on the heart failure data. We learn that (1) Black and Asian patients tend to have higher disease severity than White patients; (2) our model learns precise descriptions of health disparities and finds that disparities of multiple types exist in our setting; and (3) failing to account for the existing disparities meaningfully shifts severity estimates for all racial/ethnic groups. This analysis is descriptive and does not require evaluating held-out performance, so models are fit on all available data.

**Black and Asian patients have higher disease severity.** In Figure 4, we compare mean severity estimates for each group to the overall mean severity. Our model infers that Black and Asian patients have significantly higher disease severities than White patients ($p < 0.05$, computed by cluster bootstrapping at the patient-level).

**Model parameters capture fine-grained disparities.** As seen in Figure 3 (right), our model infers that Black and Asian patients first visit healthcare providers for heart failure significantly later in their disease progression than do White patients (inferred average initial severity $\mu_{Z_0}$ for Black and Asian patient groups is greater than for White patients by 0.22 and 0.27, respectively). To contextualize the magnitude of these disparities, if all patients progressed at the average learned progression rate across the entire population, Black patients' first heart failure visit would occur 3.0 years later in the course of their disease progression than White patients', and Asian patients' first visit would occur 3.8 years later. We also observe that $\beta_A$ for Black patients is significantly lower than that of White patients, indicating that Black patients visit healthcare providers 10% less frequently than White patients with the same disease severity. We describe these calculations in Appendix F.

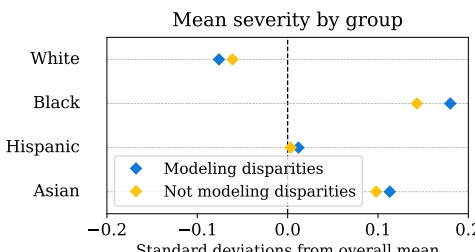 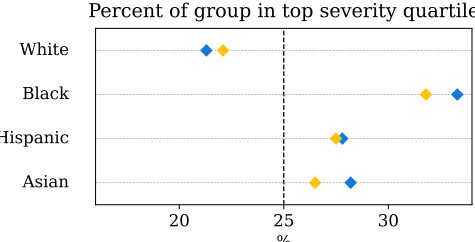

Figure 4: **Accounting for disparities leads to less biased severity estimates.** We compare the improvement of our full model (blue) over one that does not account for disparities but is otherwise the same (yellow) in two ways. On the left, we show each group's average difference from the overall mean severity, normalized by the overall standard deviation of severity. On the right, we capture the portion of each group that is identified as "high-risk" (top quartile of disease severity).

**Accounting for disparities increases estimated severity for non-white patient groups.** To assess whether accounting for disparities meaningfully shifts severity estimates, we compare severity estimates from our model to those of an ablated version of our model that does not account for disparities (but is otherwise identical). This meaningfully shifts severity estimates (Figure 4 left): while both models learn that non-white patients tend to have higher severity values, the ablated model produces higher severity estimates for White patients and lower severity estimates for all other groups ($p < 0.001$ for all groups, computed by cluster bootstrapping at the patient-level). This is consistent with our theoretical results.

To highlight some implications of these shifted severity estimates, we look at each model's ranking of patient severity levels and profile of "high-risk" patient visits: visits where inferred severity lies in the top quartile (25%) of all visits. The ablated model is significantly less likely to rank Black patient visits as high risk (Figure 4 right; $p < 0.001$, computed by cluster bootstrapping at the patient-level), skewing the demographics of the high-risk patient cohort *away* from groups that we know to have higher disease severity.

## 7 DISCUSSION

In this paper, we formalize three specific axes along which healthcare disparities emerge as biases in observed health data: underserved patients may (1) first receive care only when their disease is more severe, (2) progress faster even while receiving care, or (3) receive care less frequently even at the same disease severity. We develop a disease progression modeling approach to interpretably capture all three types of disparities while provably retaining identifiability. We prove that failing to account for any of these disparities leads to biased estimates of severity and show in a real-world heart failure dataset that accounting for health disparities does indeed meaningfully shift severity estimates by increasing the proportion of non-white patients identified as high-risk. By evaluating our model in a real healthcare setting, we validate its ability to learn fine-grained descriptions of health disparities and to make disease severity estimates that are accurate across diverse populations of patients. We thus urge future work in disease progression modeling to account for disparities in healthcare, and we lay a foundation for doing so.

There are several natural directions for future work. First, beyond heart failure, our approach could be applied to the many other progressive diseases, including Parkinson's disease (Post et al., 2005), Alzheimer's disease (Holford & Peace, 1992), diabetes (Perveen et al., 2020), and cancer (Gupta & Bar-Joseph, 2008). Second, an interesting technical direction is to extend our model to capture additional data modalities (e.g., medical images) or more flexible progression models (e.g., non-linear trajectories) while retaining its provable identifiability. Finally, our approach generalizes naturally to progression model settings beyond healthcare where disparities are of interest, including infrastructure deterioration (Madanat et al., 1995) and human aging (Pierson et al., 2019); these would be interesting domains for future work.

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

# A PROOF OF IDENTIFIABILITY

## A.1 PROOF OF THEOREM 4.1

**Theorem 4.1.** *All model parameters are identified by the observed data distribution $P(X_t, D_t \mid A)$.*

*Proof.* We want to show that each unique set of parameter assignments leads to a different distribution over the observed data. To do this, we divide our argument into four lemmas:

**Lemma A.1.** *Parameters $F, b, \Psi$ are identified by $P(X_t \mid A = a_0)$.*

*Proof.* We want to show that if two parameter sets $\{F, b, \Psi\}$ and $\{\tilde{F}, \tilde{b}, \tilde{\Psi}\}$ yield the same observed data distribution $P(X_0 \mid A = a_0)$, the parameter sets must be identical.

We first note that at $t = 0$, we have $Z_t = Z_0 \sim \mathcal{N}(0, 1)$ for group $a_0$. Then the mapping between severity and features

$$X_0 = F \cdot Z_0 + b + \epsilon_t$$

$$\epsilon_t \sim \mathcal{N}(0, \Psi)$$

captures a factor analysis model with factor loading matrix $F$ and diagonal covariance matrix $\Psi$. At $t = 0$, the feature distribution for group $a_0$ has the standard factor analysis distribution (Shapiro, 1985):

$$X_0 \sim \mathcal{N}(b, FF^T + \Psi).$$

Assuming the two sets of parameters map to distributions of $X_0$ with the same mean, it must hold that $b = \tilde{b}$. Thus, parameter $b$ is identified by data distribution $P(X_0 \mid A = a_0)$.

Further, the covariance matrix of $X_0$ induced by each set of parameters must be the same: $F(F)^T + \Psi = \tilde{F}(\tilde{F})^T + \tilde{\Psi}$. Element-wise equality of the covariance matrix gives us the following, where subscripts $i$ refer to the $i$-th element of each parameter vector:

$$F_i F_j = \tilde{F}_i \tilde{F}_j \ \ \forall i, j, i \neq j \tag{1}$$

$$(F_i)^2 + \Psi_i = (\tilde{F}_i)^2 + \tilde{\Psi}_i \tag{2}$$

Using the equality constraint (1) for multiple pairs of indices, we have that for all assignments of distinct indices $i, j, k$:

$$(F_i F_j = \tilde{F}_i \tilde{F}_j) \wedge (F_j F_k = \tilde{F}_j \tilde{F}_k) \implies \frac{\tilde{F}_i}{F_i} = \frac{\tilde{F}_k}{F_k} \tag{3}$$

$$F_i F_k = \tilde{F}_i \tilde{F}_k \implies \frac{F_i}{\tilde{F}_i} = \frac{\tilde{F}_k}{F_k} \tag{4}$$

Together, equations 3 and 4 give us:

$$\frac{\tilde{F}_i}{F_i} = \frac{F_i}{\tilde{F}_i} \implies (\tilde{F}_i)^2 = (F_i)^2 \implies F_i = \alpha \tilde{F}_i$$

where $\alpha \in \{-1, +1\}$. Since we have fixed $F_0 > 0$ for *all* factor loading matrices $F$, the sign of $\alpha$ is fixed:

$$F_0 = \alpha \tilde{F}_0 \implies \alpha = 1 \implies F_i = \tilde{F}_i \ \ \forall i \in [0, d), \tag{5}$$

meaning we have identified $F$.

Lastly, using equations (2) and (5) we get $F_i = \tilde{F}_i \implies \Psi_i = \tilde{\Psi}_i$. We have now shown that if two parameter sets induce the same distribution of $X$ at time $t = 0$, they must have the same exact value assignments. Therefore $F, b, \Psi$ are identified by $P(X_t \mid A = a_0)$. $\qquad\square$

**Lemma A.2.** *Global parameters $F, b, \Psi$ and parameters $\mu_{Z_0}^{(a)}, \sigma_{Z_0}^{(a)}, \mu_R^{(a)}, \sigma_R^{(a)}$ for each group $a$ are identified by $P(X_t \mid A)$.*

*Proof.* By Lemma A.1, we know that $F, b, \Psi$ are identified by $P(X_0 \mid A = a_0)$. We want to show that for any group $a$, if two parameter sets $\{\mu_{Z_0}^{(a)}, \sigma_{Z_0}^{(a)}, \mu_R^{(a)}, \sigma_R^{(a)}\}$ and $\{\tilde{\mu}_{Z_0}^{(a)}, \tilde{\sigma}_{Z_0}^{(a)}, \tilde{\mu}_R^{(a)}, \tilde{\sigma}_R^{(a)}\}$ yield the same observed data distribution $P(X_t \mid A = a)$, the parameter sets must be identical. In this proof we consider an arbitrary group $a$ and omit the $(a)$ superscript for brevity.

We model the following:

$$Z_0 \sim \mathcal{N}\left(\mu_{Z_0}, \sigma_{Z_0}{}^2\right)$$

$$R \sim \mathcal{N}\left(\mu_R, \sigma_R{}^2\right)$$

$$Z_t = Z_0 + R \cdot t \implies Z_t \sim \mathcal{N}\left(\mu_R \cdot t + \mu_{Z_0}, \sigma_R{}^2 \cdot t^2 + \sigma_{Z_0}{}^2\right)$$

$$X_t = F \cdot Z_t + b + \epsilon_t, \text{ where } \epsilon_t \sim \mathcal{N}(0, \Psi) \tag{6}$$

We see that equation (6) captures a factor analysis model with factor loading matrix $F$ and diagonal covariance matrix $\Psi$, meaning

$$X_t \sim \mathcal{N}(b + F(\mu_R \cdot t + \mu_{Z_0}), F(\sigma_R{}^2 \cdot t^2 + \sigma_{Z_0}{}^2)F^T + \Psi).$$

Recalling that $F_0 > 0$, we first consider $t = 0$, where $X_0 \sim \mathcal{N}(b + F\mu_{Z_0}, F(\sigma_{Z_0}{}^2)F^T + \Psi)$. In order for the two parameter sets to map to distributions of $X_0$ with the same mean, it must be the case that

$$b + F\mu_{Z_0} = b + F\tilde{\mu}_{Z_0} \implies \mu_{Z_0} = \tilde{\mu}_{Z_0}.$$

Further, for the two parameter sets to map to distributions with the same covariance matrix, it must hold that

$$F(\sigma_{Z_0}{}^2)F^T + \Psi = F(\tilde{\sigma}_{Z_0}{}^2)F^T + \Psi \implies \sigma_{Z_0} = \tilde{\sigma}_{Z_0}$$

since we know $\sigma_{Z_0}, \tilde{\sigma}_{Z_0} > 0$. So we have identified $\mu_{Z_0}$ and $\sigma_{Z_0}$. We next consider any time $t \neq 0$. For the two parameter sets to map to distributions of $X_t$ with the same mean, given that we have already shown $\mu_{Z_0}$ must equal $\tilde{\mu}_{Z_0}$, it must hold that

$$b + F(\mu_R \cdot t + \mu_{Z_0}) = b + F(\tilde{\mu}_R \cdot t + \tilde{\mu}_{Z_0}) \implies \mu_R = \tilde{\mu}_R.$$

For the two parameter sets to map to distributions with the same covariance matrix, given that we have already shown $\sigma_{Z_0}$ must equal $\tilde{\sigma}_{Z_0}$, it must hold that

$$F(\sigma_R{}^2 \cdot t^2 + \sigma_{Z_0}{}^2)F^T + \Psi = F(\tilde{\sigma}_R{}^2 \cdot t^2 + \tilde{\sigma}_{Z_0}{}^2)F^T + \Psi \implies \sigma_R = \tilde{\sigma}_R$$

since $\sigma_R, \tilde{\sigma}_R > 0$. Thus we have shown that for any group $a$, group-specific values of $\mu_{Z_0}, \sigma_{Z_0}, \mu_R, \sigma_R$ are identified by $P(X_t \mid A = a)$.

$\qquad\square$

**Lemma A.3.** *Global parameters $\beta_0, \beta_Z$ and the parameter $\beta_A^{(a)}$ for each group $a$ are identified by $P(D_t \mid A)$.*

*Proof.* We want to show that if two parameter sets $\{\beta_0, \beta_Z, \beta_A^{(a)}\}$ and $\{\tilde{\beta}_0, \tilde{\beta}_Z, \tilde{\beta}_A^{(a)}\}$ yield the same observed data distribution $P(D_t \mid A = a)$, the parameter sets must be identical. Unless otherwise specified, we consider an arbitrary group $a$ and omit the $(a)$ superscript for brevity. We also assume $\mu_R \neq 0$, since in general the severity of a progressive disease should change over time and it does not make sense to learn progression in the case that it does not.

Each event when a patient visits the hospital ($D_t = 1$) is generated by an inhomogeneous Poisson process parameterized by $\lambda_t$, where $\log(\lambda_t) = \beta_0 + \beta_Z \cdot Z_t + \beta_A$.

In order for two data distributions to have identical $P(D_t \mid A = a)$ they must have identical expected rates $\mathbb{E}_{Z_0, R}[\lambda_t]$: $\mathbb{E}_{Z_0, R}[\lambda_t]$ is the expected rate of events (across the population) at time $t$—if two distributions have a different expected rate of events at any time $t$, then $P(D_t \mid A = a_0)$ must differ at that point in time as well. Thus if two sets of parameters $\{\beta_0, \beta_Z, \beta_A\}$ and $\{\tilde{\beta}_0, \tilde{\beta}_Z, \tilde{\beta}_A\}$ yield the same observed data distribution $P(D_t \mid A = a)$, they must also generate the same observed values $\mathbb{E}_{Z_0, R}[\lambda_t]$ at all timesteps $t$. We finish the proof by showing that this holds only if $\{\beta_0, \beta_Z, \beta_A\} = \{\tilde{\beta}_0, \tilde{\beta}_Z, \tilde{\beta}_A\}$.

$$\mathbb{E}_{Z_0, R}[\lambda_t] = \int \int \lambda_t \cdot P(Z_0) \cdot P(R) \, dZ_0 dR$$

By Lemma A.2, we know that $\mu_{Z_0}, \sigma_{Z_0}, \mu_R, \sigma_R$ are identified by $P(X_t \mid A)$. Then

$$P(Z_0) = \frac{1}{\sqrt{2\pi(\sigma_{Z_0})^2}} \exp\left(-\frac{(Z_0 - \mu_{Z_0})^2}{2(\sigma_{Z_0})^2}\right)$$

$$P(R) = \frac{1}{\sqrt{2\pi(\sigma_R)^2}} \exp\left(-\frac{(R - \mu_R)^2}{2(\sigma_R)^2}\right)$$

$$\mathbb{E}_{Z_0, R}[\lambda_t] = \exp(f(\beta_0, \beta_Z, \beta_A, t)) \qquad (7)$$

where $f(\beta_0, \beta_Z, \beta_A, t) = \left(\frac{(\beta_Z \sigma_R)^2}{2}\right) t^2 + (\beta_Z \mu_R) t + \left(\beta_0 + \frac{(\beta_Z \sigma_{Z_0})^2}{2} + \beta_Z \mu_{Z_0} + \beta_A\right)$

The expression in 7 must be equal for $\{\beta_0, \beta_Z, \beta_A\}$ and $\{\tilde{\beta}_0, \tilde{\beta}_Z, \tilde{\beta}_A\}$ at all timesteps $t$. Since $\exp$ is an injective function, this means that $f(\beta_0, \beta_Z, \beta_A, t) = f(\tilde{\beta}_0, \tilde{\beta}_Z, \tilde{\beta}_A, t)$ for all $t$. By equality of polynomials, each of the individual polynomial coefficients must be equal must be equal for this to hold.

We first consider the case for group $a_0$, since we pin $\beta_A^{(a_0)}$ at 0 as a reference for all other groups. Given that we have already identified $\mu_{Z_0}^{(a_0)}, \sigma_{Z_0}^{(a_0)}, \mu_R^{(a_0)}, \sigma_R^{(a_0)}$,

$$\left(\beta_0 + \frac{(\beta_Z \sigma_{Z_0})^2}{2} + \beta_Z \mu_{Z_0}\right) = \left(\tilde{\beta}_0 + \frac{(\tilde{\beta}_Z \sigma_{Z_0})^2}{2} + \tilde{\beta}_Z \mu_{Z_0}\right) \implies \beta_0 = \tilde{\beta}_0$$

Now we return to our analysis of any arbitrary group $a$. Given that we have already identified $\mu_{Z_0}, \sigma_{Z_0}, \mu_R \neq 0, \sigma_R$,

$$\beta_Z \mu_R = \tilde{\beta}_Z \mu_R \implies \beta_Z = \tilde{\beta}_Z$$

$$\left(\beta_0 + \frac{(\beta_Z \sigma_{Z_0})^2}{2} + \beta_Z \mu_{Z_0} + \beta_A\right) = \left(\tilde{\beta}_0 + \frac{(\tilde{\beta}_Z \sigma_{Z_0})^2}{2} + \tilde{\beta}_Z \mu_{Z_0} + \tilde{\beta}_A\right) \implies \beta_A = \tilde{\beta}_A$$

Thus we have shown that $\beta_0, \beta_Z$, and $\beta_A^{(a)}$ for any group $a$ are identified by $P(D_t \mid Z_t, A)$.

$\square$

By showing that each parameter of the model is uniquely recovered from the observed data, we have proved that our model is identifiable.

$\square$

## B  PROOFS OF BIAS

In this section, in order to capture the effect of failing to account for one disparity at a time, we consider the setting where everything between two groups is the same except for disparity of focus. It is clear to see from our analysis that these results hold even more generally—as long as all existing disparities disfavor or favor the same group (e.g. a disadvantaged group with respect to one disparity is not advantaged with respect to another, in which case the effects could cancel each other out), our proofs of bias will hold. Throughout our proofs, we assume that all PDFs and conditional PDFs have positive support over their entire domain, and that all PDFs are differentiable, a very reasonable assumption over our setting.

### B.1  THEOREM 4.3

**Theorem 4.3.** *A model that does not take into account disparities in initial disease severity $Z_0$ will underestimate the disease severity of groups with higher initial severity and overestimate that of groups with lower initial severity. Specifically, if $P(Z_0 \mid A = a)$ strictly MLRPs $P(Z_0)$ for some group $a$, then $\mathbb{E}[Z_t \mid X_t] < \mathbb{E}[Z_t \mid X_t, A = a]$. Similarly, if $P(Z_0)$ strictly MLRPs $P(Z_0 \mid A = a)$ for some group $a$, then $\mathbb{E}[Z_t \mid X_t] > \mathbb{E}[Z_t \mid X_t, A = a]$.*

*Proof.* We want to show that $\mathbb{E}[Z_t \mid X_t, A = a] > \mathbb{E}[Z_t \mid X_t]$. We first show that $P(Z_0 \mid X_t = x, A = a)$ strictly MLRPs $P(Z_0 \mid X_t)$ with respect to $Z_0$:

$$
\frac{\partial}{\partial Z_0} \left( \frac{P(Z_0 \mid X_t, A = a)}{P(Z_0 \mid X_t)} \right) = \frac{\partial}{\partial Z_0} \left( \frac{\frac{P(X_t \mid Z_0, A = a) P(Z_0 \mid A = a)}{P(X_t \mid A = a)}}{\frac{P(X_t \mid Z_0) P(Z_0)}{P(X_t)}} \right) \quad \text{(Bayes Rule)}
$$

$$
= \frac{\partial}{\partial Z_0} \left( \frac{\frac{P(Z_0 \mid A = a)}{P(X_t \mid A = a)}}{\frac{P(Z_0)}{P(X_t)}} \right) \quad (X_t \perp A \mid Z_0, R)
$$

$$
= \frac{P(X_t)}{P(X_t \mid A = a)} \cdot \frac{\partial}{\partial Z_0} \left( \frac{P(Z_0 \mid A = a)}{P(Z_0)} \right)
$$

$$
> 0 \quad \text{(Disparity assumption)}
$$

Since MLRP implies first-order stochastic dominance (FOSD) (Klemens, 2007), this proves that $P(Z_0 \mid X_t, A = a)$ strictly FOSDs $P(Z_0 \mid X_t)$ and thus that $\mathbb{E}[Z_0 \mid X_t, A = a] > \mathbb{E}[Z_0 \mid X_t]$. By linearity of expectation,

$$
\mathbb{E}[Z_0 \mid X_t, A = a] + \mathbb{E}[f(R, t) \mid X_t, A = a] > \mathbb{E}[Z_0 \mid X_t] + \mathbb{E}[f(R, t) \mid X_t], \quad \forall t \geq 0
$$
$$
\implies \mathbb{E}[Z_t \mid X_t, A = a] > \mathbb{E}[Z_t \mid X_t]
$$

It is clear to see that this argument extends naturally to show that if a group tends to come in at *earlier* disease stages than the rest of the population, that their severity will be overestimated: If there exists a group $\tilde{a}$ such that $P(Z_0)$ strictly MLRPs $P(Z_0 \mid A = \tilde{a})$ with respect to $Z_0$ and $\mathbb{E}[R \mid X_t] \geq \mathbb{E}[R \mid X_t, A = \tilde{a}]$, then we will see that $\mathbb{E}[Z_t \mid X_t, A = \tilde{a}] < \mathbb{E}[Z_t \mid X_t]$. Hence any model that does not take into account demographic disparities in initial disease severity levels at a patient's first visit will lead to biased estimates of severity. $\square$

### B.2  PROOF OF THEOREM 4.4

**Theorem 4.4.** *A model that does not take into account disparities in rate of progression $R$ will underestimate the disease severity of groups with higher progression rates and overestimate that of groups with lower progression rates. Specifically, if $P(R \mid A = a)$ strictly MLRPs $P(R)$ for some group $a$, then $\mathbb{E}[Z_t \mid X_t] < \mathbb{E}[Z_t \mid X_t, A = a]$. Similarly, if $P(R)$ strictly MLRPs $P(R \mid A = a)$ for some group $a$, then $\mathbb{E}[Z_t \mid X_t] > \mathbb{E}[Z_t \mid X_t, A = a]$.*

$R$ is a patient's linear rate of progression, so we model a patient's severity over time as $Z_t = f(R, t) + Z_0$, where $f$ is linearly increasing in $R$.

*Proof.* We want to show that $\mathbb{E}[Z_t \mid X_t, A = a] > \mathbb{E}[Z_t \mid X_t]$. We first show that $P(R \mid X_t, A = a)$ strictly MLRPs $P(R \mid X_t)$ with respect to $R$:

$$\frac{\partial}{\partial R}\left(\frac{P(R \mid X_t, A = a)}{P(R \mid X_t)}\right) = \frac{\partial}{\partial R}\left(\frac{\frac{P(X_t \mid R, A=a)P(R \mid A=a)}{P(X_t \mid A=a)}}{\frac{P(X_t \mid R)P(Z_t = z_t)}{P(X_t)}}\right) \qquad \text{(Bayes Rule)}$$

$$= \frac{\partial}{\partial R}\left(\frac{\frac{P(R \mid A=a)}{P(X_t \mid A=a)}}{\frac{P(R)}{P(X_t)}}\right) \qquad (X \perp A \mid Z_0, R)$$

$$= \frac{P(X_t)}{P(X_t \mid A = a)} \cdot \frac{\partial}{\partial R}\left(\frac{P(R \mid A = a)}{P(R)}\right)$$

$$> 0 \qquad \text{(Disparity assumption)}$$

Since MLRP implies FOSD (Klemens, 2007), this also implies that $P(R \mid X_t, A = a)$ strictly FOSDs $P(R \mid X_t)$. It follows directly that $\mathbb{E}[R \mid X_t, A = a] > \mathbb{E}[R \mid X_t]$. By linearity of expectation,

$$\mathbb{E}[f(R, t) + Z_0 \mid X_t, A = a] > \mathbb{E}[f(R, t) + Z_0 \mid X_t], \quad \forall t > 0$$
$$\implies \mathbb{E}[Z_t \mid X_t, A = a] > \mathbb{E}[Z_t \mid X_t]$$

It is clear to see that this argument extends naturally to show that if a group tends to progress *more slowly* than the rest of the population, that their severity will be overestimated: if there exists a group $\tilde{a}$ such that $P(R)$ strictly MLRPs $P(R \mid A = \tilde{a})$ with respect to $R$ and $\mathbb{E}[Z_0 \mid X_t] \geq \mathbb{E}[Z_0 \mid X_t, A = \tilde{a}]$, then we will see that $\mathbb{E}[Z_t \mid X_t, A = \tilde{a}] < \mathbb{E}[Z_t \mid X_t]$. Thus any model that does not take into account demographic disparities in patient progression rates will lead to biased estimates of severity. $\qquad\square$

### B.3 PROOF OF THEOREM 4.5

**Theorem 4.5.** *A model that does not take into account disparities in visit frequency $\lambda_t$ (conditional on disease severity) will underestimate the disease severity of groups with lower visit frequency and overestimate that of groups with higher visit frequency. Specifically, if it holds for some group $a$ that $\beta_A^{(a)} < \beta_A^{(\tilde{a})}$ for all $\tilde{a} \neq a$, then $\mathbb{E}[Z_t \mid D_t] < \mathbb{E}[Z_t \mid D_t, A = a]$. Similarly, if it holds for some group $a$ that $\beta_A^{(a)} > \beta_A^{(\tilde{a})}$ for all $\tilde{a} \neq a$, then $\mathbb{E}[Z_t \mid D_t] > \mathbb{E}[Z_t \mid D_t, A = a]$.*

We model a patient's visit pattern using an inhomogeneous poisson process characterized by visit rate $\lambda_t$, such that $\log(\lambda_t) = g(Z_t) + \beta_A^{(A)}$ for some function of severity $g(Z_t)$ and group-specific adjustments $\beta_A^{(A)}$. In our proof, we assume the large-sample limit in which $\lambda_t$ can be perfectly estimated from the observed data, and thus treat it as observed; we show empirically that our results hold in finite samples as well. We assume $g(Z_t)$ is a strictly monotonically increasing function of severity.

*Proof.* We want to show that $\mathbb{E}[Z_t \mid D_t, A = a] > \mathbb{E}[Z_t \mid D_t]$. We do this by calculating each term separately.

We first consider $\mathbb{E}[Z_t \mid D_t, A = a]$. Observing $D_t$ over time gives us an observed value of visit rate $\lambda_t$. The strictly monotone assumption of $g$ ensures $g$ is invertible, and the fact that all visit rates $\lambda_t$ are characterized by $\log(\lambda_t) = g(Z_t) + \beta_A^{(A)}$ ensures that this holds over the entire range of $\lambda_t$ values. This gives us:

$$\mathbb{E}[Z_t \mid D_t, A = a] = \mathbb{E}\left[g^{-1}\left(\log(\lambda_t) - \beta_A^{(A)}\right) \;\middle|\; D_t, A = a\right]$$

$$= g^{-1}\left(\log(\lambda_t) - \beta_A^{(a)}\right)$$

We next consider the case where a model infers severity without taking into account disparities in visit rate conditional on severity. Estimating severity $Z_t$ based solely on visit observations gives:

$$\mathbb{E}[Z_t \mid D_t] = P(A = a) \cdot \mathbb{E}[Z_t \mid D_t, A = a] + P(A \neq a) \cdot \mathbb{E}[Z_t \mid D_t, A \neq a]$$

$$= P(A = a) \cdot \mathbb{E}\left[g^{-1}\left(\log(\lambda_t) - \beta_A^{(A)}\right) \mid D_t, A = a\right]$$

$$+ P(A \neq a) \cdot \mathbb{E}\left[g^{-1}\left(\log(\lambda_t) - \beta_A^{(A)}\right) \mid D_t, A \neq a\right]$$

$$< P(A = a) \cdot \mathbb{E}\left[g^{-1}\left(\log(\lambda_t) - \beta_A^{(A)}\right) \mid D_t, A = a\right]$$

$$+ P(A \neq a) \cdot \mathbb{E}\left[g^{-1}\left(\log(\lambda_t) - \beta_A^{(a)}\right) \mid D_t, A = a\right] \qquad (*)$$

$$= P(A = a) \cdot \left(g^{-1}\left(\log(\lambda_t) - \beta_A^{(a)}\right)\right) + P(A \neq a) \cdot \left(g^{-1}\left(\log(\lambda_t) - \beta_A^{(a)}\right)\right)$$

$$= g^{-1}\left(\log(\lambda_t) - \beta_A^{(a)}\right)$$

$$= \mathbb{E}[Z_t \mid D_t, A = a]$$

As justification for $(*)$:

$$\beta_A^{(a)} < \beta_A^{(A)}, \quad \forall A \neq a \qquad \text{(Disparity assumption)}$$

$$\implies \log(\lambda_t) - \beta_A^{(a)} > \log(\lambda_t) - \beta_A^{(A)}, \quad \forall A \neq a, \forall \lambda_t$$

$$\implies g^{-1}\left(\log(\lambda_t) - \beta_A^{(a)}\right) > g^{-1}\left(\log(\lambda_t) - \beta_A^{(A)}\right), \quad \forall A \neq a, \forall \lambda_t$$

$$(g \text{ strictly monotonically increasing} \implies g^{-1} \text{ strictly monotonically increasing})$$

$$\implies \mathbb{E}\left[g^{-1}\left(\log(\lambda_t) - \beta_A^{(a)}\right) \mid D_t, A = a\right] > \mathbb{E}\left[g^{-1}\left(\log(\lambda_t) - \beta_A^{(A)}\right) \mid D_t, A \neq a\right]$$

It is clear to see that this argument extends naturally to show that if a group tends to visit the hospital *more frequently* conditional on severity, that their severity will be overestimated: if there exists a group $\tilde{a}$ such that $\beta_A^{(\tilde{a})} > \beta_A^{(A)}$ for all $A \neq \tilde{a}$, then we will see that $\mathbb{E}[Z_t \mid D_t, A = \tilde{a}] < \mathbb{E}[Z_t \mid D_t]$. Thus any model that does not take into account demographic disparities in patient visit rates given their severity will lead to biased estimates of severity. $\qquad \square$

## C  SIMULATIONS

Figure S1 shows the results of 30 simulation runs, where we randomly instantiate the parameters of our model and then generate data to fit on. We generate simulated data for 1000 patients on each run, each of whom is assigned to one group (50% chance of being from either group). We visualize the recovery of each parameter by plotting true parameter values versus recovered posterior mean values, with one dot per run.

To generate data with prevalent disparities, we set our priors to $\mu_{Z_0} \sim \mathcal{N}(0, 2.5)$ and $\sigma_{Z_0} \sim \mathcal{TN}(1, 0.5)$ (normal distribution restricted to positive values) for the non-pinned group; $\mu_R \sim \mathcal{N}(0, 3)$ and $\sigma_R \sim \mathcal{TN}(1, 0.01)$ (normal distribution restricted to positive values) for both groups; $F \sim \mathcal{TN}(1, 1)$ (normal distribution restricted to values above 0.5 to enforce positive constraint) for $F_0$; $F \sim \mathcal{N}(0, 2)$ for all other features; $b \sim \mathcal{N}(0, 1)$; $\psi \sim \mathcal{TN}(8, 1)$ (normal distribution restricted to positive values); $\beta_0 \sim \mathcal{N}(1.5, 0.1)$; $\beta_Z \sim \mathcal{N}(0.5, 0.1)$; and $\beta_A \sim \mathcal{N}(0, 2)$ for the non-pinned group.

## D  NYP HEART FAILURE DATA PROCESSING

This study was conducted in accordance with Health Insurance Portability and Accountability Act (HIPAA) guidelines and with Institutional Review Board (IRB) approval.

**Cohort filtering.**   We analyze patients with *heart failure with reduced ejection fraction* (HFrEF) whom we identify, following clinical guidance, by filtering the available NYP data for patients who have at least one LVEF measurement below 50% and who have been recorded as receiving a diuretic prescription. To ensure we have relatively complete records for each patient, we then filter for patients who are likely to receive most of their cardiology care within the NYP system, by filtering for patients whose home zipcode is in the New York metropolitan area and who have at least two LVEF or BNP records at least 6 months apart within our data. Lastly, NYP switched electronic health record (EHR) systems, introducing inconsistencies in record-keeping across sites and years; to ensure our records are consistently recorded, we analyze data from Weill Cornell Medical Center, one of NYP's two largest sites, between January 1, 2012 (the start of reliable record-keeping) to October 2, 2020 (NYP Cornell's transition to a new EHR). This ensures records are consistently recorded in our data.

**Feature processing.**   We convert pBNP to BNP with the conversion pBNP = 6.25 * BNP (Rørth et al., 2020) and then log-transform BNP values to get one combined $\log_2(\text{BNP})$ feature (Hendricks et al., 2022). We then normalize (z-score) all feature values so that each feature has mean 0 and variance 1. Because patient blood pressure and heart rate are much more likely to be measured at hospital visits unrelated to heart failure (while visiting another specialist in the NYP system), we limit patient observations to visits where a patient had one measurement of at least one of LVEF and BNP.

We encode demographic categories by making $A$ a one-hot encoding of race/ethnicity groups. Lastly, we describe the time scale of our model. As mentioned in §6, we discretize time in 1-week bins; if a patient has multiple measurements of one feature within a timestep, we average all measurements within that timestep. Discretizing time in this way allows us to capture more long-term changes rather than acute changes in patient status. We normalize time so that the total time range in our model is 0 to 1. The longest patient trajectory in our data is 446 weeks (timesteps), so we normalize timestep values so that they range from 0 to 1; we therefore have fractional, discrete time values, each representing one week as $1/446$ units of time.

## E   MODEL EVALUATION

**Fitting model on real data.**   We fit our model on real data using weakly informative priors: $\mu_{Z_0} \sim \mathcal{N}(0, 1)$ and $\sigma_{Z_0} \sim \mathcal{TN}(1, 1)$ (normal distribution restricted to positive values) for the non-pinned groups; $\mu_R \sim \mathcal{N}(0, 1)$ and $\sigma_R \sim \mathcal{TN}(1.5, 1)$ (normal distribution restricted to positive values) for both groups; $b \sim \mathcal{N}(0, 1)$; $\psi \sim \mathcal{TN}(1, 0.5)$ (normal distribution restricted to positive values); $\beta_0 \sim \mathcal{N}(2.5, 1)$; $\beta_Z \sim \mathcal{N}(0, 1)$; and $\beta_A \sim \mathcal{N}(0, 1)$ for the non-pinned group.

For $F$, we set model priors using Factor Analysis: at $t = 0$, we have $Z_t = Z_0 \sim \mathcal{N}(0, 1)$ for group $a_0$, meaning the mapping between severity and features

$$X_0 = F \cdot Z_0 + b + \epsilon_t$$
$$\epsilon_t \sim \mathcal{N}(0, \Psi)$$

captures a factor analysis model with factor loading matrix $F$ and diagonal covariance matrix $\Psi$. We run factor analysis using feature measurements from the *first timestep* of all White patients (our $a_0$ group) and use the estimates of $F$ from Factor Analysis as the mean of our priors on $F$. We define the variance of our priors on $F$ to be 1, and we pin the sign of $F_{\text{LVEF}}$ to be negative for identifiability. Since we have no inherent value scale for what $F$ values should be, Factor Analysis allows us to fit the model on more substantiated priors for feature scaling factors.

We then fit the model and get the parameter estimates from 1000 samples. For any time $t$, we can calculate an estimate of $Z_t$ and $X_t$ for each sample, based on the sample's parameter estimates; we then take the average over all samples to get a patient's estimate of $Z_t$ and $X_t$. In order to get actual feature value estimates, we can linearly transform $X_t$ to undo the normalization for each feature and recover an estimate of each feature value at $t$. We can then use our model's estimates of $Z_t$ and predicted feature values to analyze and evaluate our model's behavior.

**Comparison to baselines.**   We filter out patients who do not have at least three visits (since several of the baselines we fit require this many visits per patient, as we describe below), leaving a total of 1834 patients: 1118 White, 347 Black, 216 Hispanic, and 153 Asian patients.

To evaluate our model's ability to reconstruct feature values, we compare our model to PCA and FA. PCA and FA require consistent dimensionality of the input data, so we fit all models on the first three visits for each patient. We train two variants of both PCA and FA: the first attempts to reconstruct patient *visits* from a single latent dimension (analogous to $Z$ in our model), taking as input the $X_t$ vector at one visit (4 features total) and representing it with a single latent component. The second variant attempts to reconstruct *patient trajectories* from two latent dimensions (analogous to $Z_0$ and $R$ in our model), taking as input a concatenated vector of features $X_t$ from the first three visits (12 features total) and representing it with two latent components. We impute missing values as the overall mean of the data for both PCA and FA, since these methods cannot naturally handle missing data.

To evaluate our model's ability to predict future feature values, we compare our model to last time-step, logistic regression, and quadratic regression. Unlike PCA and FA, these methods do not require consistent dimensionality in the input data, so we fit the models to the first three years of observed data. Last-timestep predicts all future feature values to be equal to the most recent feature value observed in the training data for that patient; if there is no observed feature value, the baseline predicts the population mean. Linear regression regresses values on time for each patient and each feature to predict future feature values. For patients with fewer than 2 observations for a given feature value, we use the population mean for the preceding or subsequent timestep. Quadratic regression follows a similar approach. Because linear regression and quadratic regression can overfit the data and make unrealistic predictions, we clip their predicted feature values to a range determined by that observed within the training data.

**Ablated Model.** We compare our full model to an ablated version of the model that does not account for any of our three disparities. We do this by removing all group-specific parameters from the model, while leaving everything else the same: we learn one value of $\mu_R$ and $\sigma_R$ and exclude $\beta_A$ from the model. Since the distribution of $Z_0$ must be fixed for at least one group for identifiability (to fix the scale of $Z_t$), the distribution is pinned for all groups. Factor Analysis for model priors on $F$ is also fit on all patients rather than only on white patients.

## F  DISPARITIES ESTIMATES

We first describe our calculations for §6.3 to estimate how much later Black and Asian patients start receiving care for heart failure compared to White patients. Our model learns the following:

$$\mu_{Z_0}^{(\text{Black})} = \mu_{Z_0}^{(\text{White})} + 0.22$$
$$\mu_{Z_0}^{(\text{Asian})} = \mu_{Z_0}^{(\text{White})} + 0.27$$

The learned average rate of progression across all patients is $0.62$. This means that Black patients come in $0.22/0.62 = 0.35$ units of time later in their disease progression than White patients, and Asian patients come in $0.27/0.62 = 0.44$ units of time later than White patients. Given that one unit of time is the longest patient trajectory, $8.5$ years, this leads us to $3.0$ and $3.8$ years for Black and Asian patients, respectively.

Next we describe our calculations to estimate how much less frequently Black patients visit the hospital than White patients at the same disease severity. Our model learns that

$$\beta_A^{(\text{Black})} = \beta_A^{(\text{White})} - 0.11$$

At the same disease severity $Z_t$, Black patients will have a visit rate of

$$\lambda_t = \exp(\beta_0 + \beta_Z \cdot Z_t + (\beta_A^{(\text{White})} - 0.11))$$
$$= \exp(\beta_0 + \beta_Z \cdot Z_t + \beta_A^{(\text{White})}) \cdot \exp(-0.11)$$
$$= 0.897 \cdot \exp(\beta_0 + \beta_Z \cdot Z_t + \beta_A^{(\text{White})})$$

So at the same disease severity, we estimate that Black patients have a visit rate that is 90% that of a White patient's visit rate.

# G SUPPLEMENTARY FIGURES AND TABLES

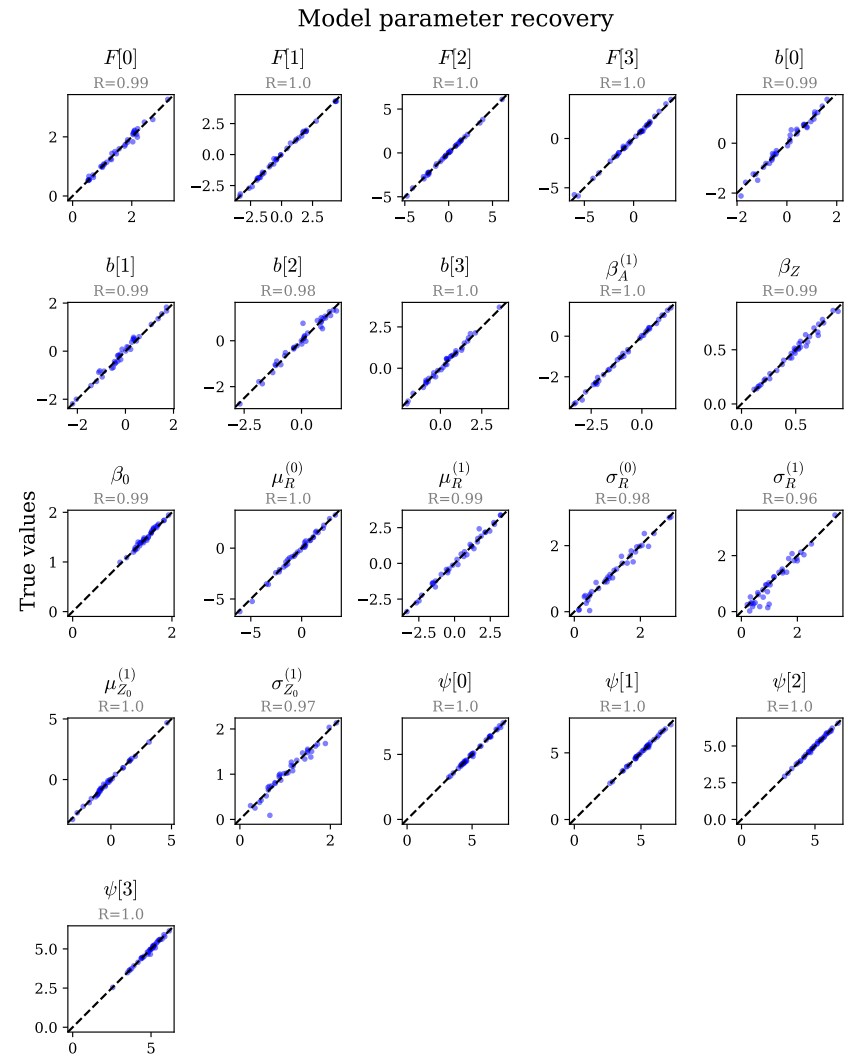

Figure S1: **Parameter recovery from fitting our model to synthetic data.** The priors from which we draw the synthetic data are: $\mu_{Z_0} \sim \mathcal{N}(0, 2)$ and $\sigma_{Z_0} \sim \mathcal{TN}(1, 1)$ (normal distribution restricted to positive values) for the non-pinned group; $\mu_R \sim \mathcal{N}(0, 2)$ and $\sigma_R \sim \mathcal{TN}(1, 1)$ (normal distribution restricted to positive values) for both groups; $F \sim \mathcal{TN}(1, 1)$ (normal distribution restricted to values above 0.5 to enforce positive constraint) for $F_0$; $F \sim \mathcal{N}(0, 2)$ for all other features; $b \sim \mathcal{N}(0, 1)$; $\psi \sim \mathcal{TN}(5, 1)$ (normal distribution restricted to positive values); $\beta_0 \sim \mathcal{N}(1.5, 0.2)$; $\beta_Z \sim \mathcal{N}(0.5, 0.1)$; and $\beta_A \sim \mathcal{N}(0, 2)$ for the non-pinned group.

|  | **Our model** | **FA$_{visit}$** | **PCA$_{visit}$** | **FA$_{patient}$** | **PCA$_{patient}$** |
|---|---|---|---|---|---|
| RMSE: informative | 0.67 | 0.86 | 0.77 | 0.76 | 0.67 |
| RMSE: all | 0.82 | 0.89 | 0.77 | 0.77 | 0.72 |

Table S1: **Our model compared to standard baselines for reconstruction performance.** We compare to factor analysis and principal component analysis fit at the patient visit level (FA$_{visit}$, PCA$_{visit}$) and at the trajectory level (FA$_{patient}$, PCA$_{patient}$). Models are fit on the first 3 visits from each patient and evaluated on same data using root mean squared error (RMSE).

|  | **Our model** | **Linear regression** | **Quadratic regression** | **Latest timestep** |
|---|---|---|---|---|
| RMSE: informative | 0.99 | 1.6 | 2.3 | 0.89 |
| RMSE: all | 0.98 | 1.8 | 2.5 | 0.98 |

Table S2: **Our model compared to standard baselines for predictive performance.** We compare to linear regression, quadratic regression, and latest timestep prediction, each fit at the patient feature level. Models are fit on data from the first 3 years of each patient's disease trajectory and evaluated on visits after 3 years using root mean squared error (RMSE).

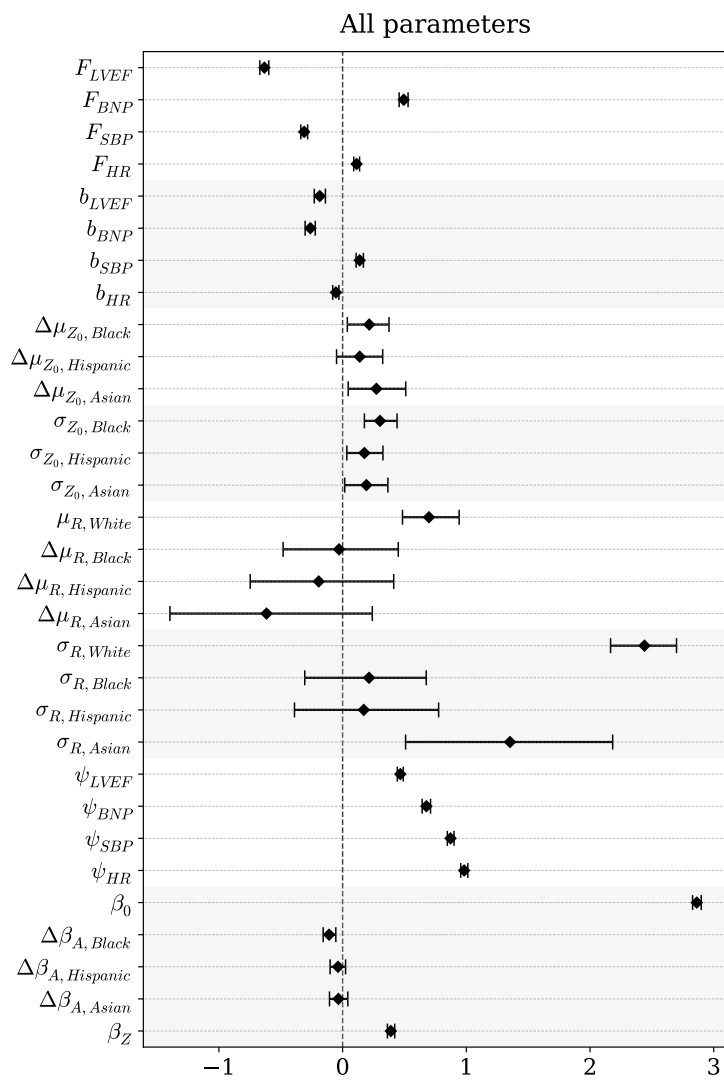

Figure S2: **All parameters learned from fitting model on NYP heart failure cohort.** Parameters of primary interest for interpreting our model (a subset of the parameters shown here) are highlighted in Figure 3.

