# OpenReview forum: "Learning Disease Progression Models That Capture Health Disparities"
_ICLR.cc/2025/Conference — ICLR 2025 Conference Withdrawn Submission_

### Official Review · Reviewer_QNCQ · 2024-10-31

**Soundness:** 3
**Presentation:** 3
**Contribution:** 3
**Rating:** 8
**Confidence:** 3

**Summary:**

This paper introduces a Bayesian modelling strategy to model disease evolution while accounting for three sources of biases present in medical data.

**Strengths:**

The paper is well written, easy to read and tackles an important problem: improving modelling when disparities mark model. The paper presents theoretical justification and thoroughly evaluates the proposed methodology on both synthetic and real-world data.

**Weaknesses:**

The model makes assumptions upon the expression of disparities while being identifiable. The underlying process must verify the assumptions. It would be beneficial to discuss further how realistic and/or common these assumptions are. An analysis of a misspecified model when the underlying generating process does not meet these assumptions would be valuable.

**Questions:**

None

---

### Official Review · Reviewer_1QRD · 2024-11-01

**Soundness:** 1
**Presentation:** 3
**Contribution:** 2
**Rating:** 3
**Confidence:** 4

**Summary:**

The authors propose a Bayesian hierarchical, linear model of disease progression that captures well known disparities in an interpretable way, while maintaining identifiability. The authors show on a simulated data set that the model can correctly identify the relevant parameters and that accounting for disparities is important to correctly estimate progression. They then demonstrate the utility on a real world data set, showing predictive ability, reconstruction compared to factor analysis and PCA and consistency with medical knowledge.

**Strengths:**

- Alleviating health disparities and fairness w.r.t. clinical algorithms is an important aspect of machine learning for health and the impact of disparities on disease progression modeling is important
- Interpretability and estimating effects of disparities and covariates on disease progression is clinically meaningful

**Weaknesses:**

- While the model is interesting from a clinical perspective, I am not sure this is the right venue for this publication due to limited technical novelty
- Very limiting disease progression trajectory due to linear assumption in time. A patient can therefore not experience worsening and improvement on their disease trajectory
- The proofs show that not taking into account disparities will bias the result, however other (non-linear) disease progression models can take "baseline" covariates like ancestry, effectively conditioning the probability of the latent progression $z$ on $A$ as well. Comparisons to models like these are necessary to underline the claims.
- The authors need to compare the disease progression modeling with other state of the art methods of inferring latent disease trajectories. It is not clear how well more realistic disease trajectories are captured and how the model compares to SOTA models on this task.
- For the synthetic data it would be helpful to also use data not sampled from the same architecture as the model under consideration, but more complex disease progressions to examine how well the model can infer these despite the relatively simple assumptions

**Questions:**

- For a continuous A, does there have to exist a reference value a_0 for which $\mu(a_0) = 0, \sigma(a_0) = 1$?
- Is the second disparity mentioned a disparity because there might be worse clinical care for each ancestry group? Or is this supposed to capture biological differences? Wouldn't the point of any disease progression model with ancestry as a covariate be to infer disparities in disease progression rates?
- What happens if some of the disparities are unmeasured? How well can the model infer the correct disease progression e.g. in simulations where some of the disparity variables are not shown to the model.

---

### Official Review · Reviewer_RYGg · 2024-11-03

**Soundness:** 3
**Presentation:** 3
**Contribution:** 3
**Rating:** 3
**Confidence:** 4

**Summary:**

In this work, the authors propose a Bayesian disease progression model that explicitly accounts for 3 types of disparities concerning health. The model contains several subgroup-specific parameters to account for inequalities in health. The authors provide a strong theoretical analysis of the model, as well as analyses of simulated data and a real application for heart failure patients.

**Strengths:**

The paper has several interesting ideas
- the addressed problem is very important
- the model is well explained, and the theoretical analysis seems strong
- the authors provide an extensive empirical analysis, with interesting insights

**Weaknesses:**

However, the paper suffers from several flaws. I am really willing to increase my grade if those points are addressed, but as it is, the contribution of the model compared to existing strategies, in terms of performance is really unclear.
- the baselines seem quite weak. The authors report that several indicators are important, like the visit frequency. I am not sure to understand from the manuscript which features the baselines include, in particular do they include demographics information? and not sure either that PCA or FA are the best tools to do feature engineering. In particular all those approaches are linear, so they can not model interactions between demographics and other features. Maybe considering tree-based methods would be relevant here, with an appropriate feature engineering process. I know that prediction is not the end-goal of the model, but this constitutes the only "measurable" performance indicator.
- the ablation studies are interesting, but they should also be conducted on the real-data application, to assess the variation in predictive performance, because it is not very convincing that data simulated with a model would be less accurately represented by a different model.
- additional baselines to consider, both on simulated and real data it would be interesting to compare the performances with models that account for demographics characteristics differently (with one type of disparity for instance)

**Questions:**

- can the authors add stronger baselines, both from classical machine learning and from the literature of disease progression with health disparities? With a better description of the feature engineering and alternative feature engineering strategies. As it is, the paper shows measurable health disparities of several types, but it is not clear that it is really important for the overall disease understanding
- can the authors extend the ablation study to the real dataset? or to datasets simulated with alternative data generation processes?
- can you describe more clearly the metrics used? there are several variables in Xt, how is the RMSE and MAPE aggregated over all of them?

---

### Official Review · Reviewer_AAFG · 2024-11-03

**Soundness:** 1
**Presentation:** 2
**Contribution:** 2
**Rating:** 3
**Confidence:** 3

**Summary:**

This paper proposes a disease progression model that uses observed symptoms to model the progression of a patient's latent severity. Compared with previous research, it accounts for three types of health disparities: initial severity, disease progression rate, and visit frequency. The proposed method is identifiable and shows good performance on a private dataset.

**Strengths:**

This paper studies the important topic of predicting disease progression by capturing the disparities between patients.

**Weaknesses:**

1. The proposed method appears to be a variant of a hidden Markov model (HMM). Instead of using transition probabilities in HMM, it employs simple functions to describe transitions between states and outcomes. This simplification might limit the model's ability to capture the complex dynamics of disease progression.
2. The directed acyclic graph, the selection of functions between observed and hidden variables, and the specific types of disparities incorporated in Section 3 seem overly simplistic and heuristic. The paper lacks detailed reasoning or motivation for these design choices. Providing justifications or empirical evidence supporting these decisions would enhance the credibility of the model.
3. The paper adopts a linear representation solely because "it provides an interpretable characterization of the trajectory." However, real-world disease progression often involves intricate, nonlinear relationships between variables. Relying solely on a linear model may lead to suboptimal performance and may not capture the true underlying patterns, potentially undermining the trustworthiness of the explanations. Exploring nonlinear models could offer better performance and more reliable interpretations.
4. The empirical results are based on a private dataset with a relatively small number of subjects (n=2,942) and only four features. This raises concerns about the model's generalizability to other datasets. I recommend validating the model on public EHR datasets such as MIMIC or UK Biobank to assess its broader applicability. Furthermore, the comparisons are limited to simple baselines like linear regression, quadratic regression, principal component analysis, and factor analysis. Evaluating the model against state-of-the-art methods, including neural networks, RNNs, and transformers, would provide a more comprehensive assessment.
5. Insufficient Evidence of Superior Performance:
   - The model's explainability is only qualitatively assessed using medical knowledge. Incorporating quantitative evaluations or user studies could strengthen the claims about interpretability.
   - The model does not show (significant) improvements over the baselines in reconstruction and predictive tasks. Additionally, the absence of error bars or statistical significance tests makes it challenging to determine if the differences are meaningful. If the authors can include the code, my confidence in the results will be higher.

**Questions:**

1. Regarding function $f$ in line 135, is it a stationary function over the progression of severity -- that is, is it the same for every $Z_t$? If so, can you provide a justification for this assumption?
2. There are hundreds, if not more, types of disparities among patients. Why do you believe that the three disparities used in your paper are the most significant or important?
3.  In the methodology section, the author mentions the need to pin a group $a_0$ first. How did you choose this particular group, and does selecting a different $a_0$ affect the performance of your model?

---

### Note · Authors · 2024-11-15

**Comment:**

Thank you very much for your time and feedback -- these reviews will certainly help us improve the paper. In light of the reviewer scores, we've decided to withdraw our submission, but we will take all of the suggestions into account as we revise the paper!

**Withdrawal Confirmation:**

I have read and agree with the venue's withdrawal policy on behalf of myself and my co-authors.